# Machine Learning for Early Diagnosis of ATTRv Amyloidosis in Non-Endemic Areas: A Multicenter Study from Italy

**DOI:** 10.3390/brainsci13050805

**Published:** 2023-05-16

**Authors:** Vincenzo Di Stefano, Francesco Prinzi, Marco Luigetti, Massimo Russo, Stefano Tozza, Paolo Alonge, Angela Romano, Maria Ausilia Sciarrone, Francesca Vitali, Anna Mazzeo, Luca Gentile, Giovanni Palumbo, Fiore Manganelli, Salvatore Vitabile, Filippo Brighina

**Affiliations:** 1Department of Biomedicine, Neuroscience and Advanced Diagnostics (BiND), University of Palermo, 90127 Palermo, Italy; vincenzo19689@gmail.com (V.D.S.); francesco.prinzi@unipa.it (F.P.); alongep95@gmail.com (P.A.); salvatore.vitabile@unipa.it (S.V.); 2Fondazione Policlinico Universitario A, Gemelli-IRCCS, UOC Neurologia, 00168 Rome, Italy; mluigetti@gmail.com (M.L.); angela.romano12@gmail.com (A.R.); a.sciarrone97@gmail.com (M.A.S.); vitali.francesca95@gmail.com (F.V.); 3Department of Neurosciences, Università Cattolica del Sacro Cuore, 00168 Rome, Italy; 4Department of Clinical and Experimental Medicine, University of Messina, 98182 Messina, Italy; russom@unime.it (M.R.); annamazzeo@yahoo.it (A.M.); luca.gentile@unime.it (L.G.); 5Department of Neuroscience, Reproductive and Odontostomatological Science, University of Naples “Federico II”, 80131 Naples, Italy; ste.tozza@gmail.com (S.T.); gio.palu1995@gmail.com (G.P.); fioremanganelli@gmail.com (F.M.)

**Keywords:** TTR, hereditary amyloid neuropathy, genetic screening, ATTRv, machine learning, genetic testing

## Abstract

Background: Hereditary transthyretin amyloidosis with polyneuropathy (ATTRv) is an adult-onset multisystemic disease, affecting the peripheral nerves, heart, gastrointestinal tract, eyes, and kidneys. Nowadays, several treatment options are available; thus, avoiding misdiagnosis is crucial to starting therapy in early disease stages. However, clinical diagnosis may be difficult, as the disease may present with unspecific symptoms and signs. We hypothesize that the diagnostic process may benefit from the use of machine learning (ML). Methods: 397 patients referring to neuromuscular clinics in 4 centers from the south of Italy with neuropathy and at least 1 more red flag, as well as undergoing genetic testing for ATTRv, were considered. Then, only probands were considered for analysis. Hence, a cohort of 184 patients, 93 with positive and 91 (age- and sex-matched) with negative genetics, was considered for the classification task. The XGBoost (XGB) algorithm was trained to classify positive and negative *TTR* mutation patients. The SHAP method was used as an explainable artificial intelligence algorithm to interpret the model findings. Results: diabetes, gender, unexplained weight loss, cardiomyopathy, bilateral carpal tunnel syndrome (CTS), ocular symptoms, autonomic symptoms, ataxia, renal dysfunction, lumbar canal stenosis, and history of autoimmunity were used for the model training. The XGB model showed an accuracy of 0.707 ± 0.101, a sensitivity of 0.712 ± 0.147, a specificity of 0.704 ± 0.150, and an AUC-ROC of 0.752 ± 0.107. Using the SHAP explanation, it was confirmed that unexplained weight loss, gastrointestinal symptoms, and cardiomyopathy showed a significant association with the genetic diagnosis of ATTRv, while bilateral CTS, diabetes, autoimmunity, and ocular and renal involvement were associated with a negative genetic test. Conclusions: Our data show that ML might potentially be a useful instrument to identify patients with neuropathy that should undergo genetic testing for ATTRv. Unexplained weight loss and cardiomyopathy are relevant red flags in ATTRv in the south of Italy. Further studies are needed to confirm these findings.

## 1. Introduction

Hereditary transthyretin amyloidosis with polyneuropathy (ATTRv) is an adult-onset, rare, and multisystemic disease, affecting the sensorimotor and autonomic functions along with other organs, especially the heart, gastrointestinal tract, eyes, and kidneys [1]. ATTRv is caused by the accumulation of abnormal amyloid fibrils originating from mutations in the *TTR* gene; ATTRv displays incomplete penetrance and presents an autosomal dominant pattern of inheritance [2,3]. The clinical phenotype is heterogeneous and often unpredictable; therefore, the diagnosis is very difficult and, in most cases, delayed [4]. Multisystemic clinical presentation of ATTRv often makes it difficult to distinguish ATTRv from other conditions, thus causing a significant misdiagnosis [5,6]. For example, chronic inflammatory demyelinating polyradiculoneuropathy, diabetes, sensory ataxia, and amyotrophic lateral sclerosis (ALS) commonly overlap with ATTRv [5,6,7]. Unfortunately, misdiagnosis of ATTRv carries high costs for the community in terms of mortality and inappropriate treatments [8], because several treatment options are available if the correct diagnosis is achieved, and these treatments are particularly effective in early disease stages [9,10,11]. Hence, avoiding misdiagnosis is crucial [4]. In the past, the diagnosis of ATTRv required genetic testing performed upon a strong clinical suspicion in the presence of a positive biopsy [8,12]. More recently, the role of a biopsy has no longer become irreplaceable due to the broad availability of genetic testing [13,14]. Indeed, more recent diagnostic algorithms suggest anticipating and often replacing the biopsy in the diagnostic workup [1]. In addition, the presence of a positive family history is not always present in ATTRv patients, and it is not infrequent for clinicians to face difficult cases with a sporadic onset, due to the wide heterogeneity within families and incomplete penetrance [15]. Consequently, based on the published literature and expert opinions, symptom clusters and specific “red flags” have recently been proposed to facilitate an earlier diagnosis [4,15]. However, there is still a need for new strategies to find undiagnosed individuals and implement existing evidence-based guidelines to improve ATTRv care [12].

Machine learning (ML) algorithms have shown remarkable capabilities for the development of classification models. In particular, their strength lies in finding hidden relationships among the variables to predict a clinical outcome. In fact, recently, it has been reported that data mining can improve the prediction and diagnostic precision in several different conditions including cardiac amyloidosis [16,17,18]. However, despite the fact that they enable the training of high-accuracy models, many ML algorithms suffer from the problem of low interpretability and transparency [19]. The lack of transparency prompts general skepticism of these new technologies and complicates their integration into clinical practice [20,21]. In recent years, to interpret and explain the developed models, the training of ML classifiers is followed by explainable AI (XAI) algorithms [22]. The use of XAI algorithms allows for an understanding of the most important features involved in the prediction process and for a comparison of model findings with the medical literature. In addition, it is possible to avoid several issues that could affect the model’s reliability and detect abnormal behaviors and erroneous findings.

We hypothesize that the use of ML and XAI methods in the genetic screening for ATTRv might lead to a higher sensitive and specific diagnostic approach, thus contributing to a significant reduction in the diagnostic delay of ATTRv in non-endemic areas, as well as ensuring the early treatment for this rare inherited disease. This study aims to evaluate the role of machine learning algorithms in the prediction of ATTRv, diagnosed by means of genetic testing.

## 2. Methods

### 2.1. Study Procedures

This study was approved by the Ethical Committee of Messina on 22 March 2016 (V n.3/2016), and it was conducted in conformity with the Declaration of Helsinki principles. Patients suspected of having ATTRv based on specific “red flags” [4,15] were enrolled. In a second phase, they went through a complete diagnostic workup for ATTRv including genetic testing. Clinical data have been retrospectively collected from patients undergoing *TTR* genotyping in four centers specializing in the diagnosis and care of ATTRv (the neuromuscular clinics of Palermo, Messina, Naples, and Rome). For each patient undergoing genetic testing, the presence of specific “red flags” was investigated. Then, clinical data were compared in patients with positive and negative genetics; afterward, the red flags included in this study were combined to define a precise algorithm for diagnosis through a “Machine Learning” model.

### 2.2. Patient’s Population

Patients with chronic axonal sensorimotor polyneuropathy referring to the neuromuscular centers of Palermo, Rome, Messina, and Naples were retrospectively included in this study. Inclusion criteria were (1) informed consent for genetic testing; (2) age > 18 years; (3) presence of at least one red flag to raise suspicion of ATTRv. Exclusion criteria were (1) lack of informed consent; (2) no eligibility for genetic testing. Data were available for all patients followed in the four centers for ATTRv. All patients enrolled were examined with a detailed questionnaire exploring the presence of “red flags” for ATTRv and underwent genetic testing for ATTRv.

### 2.3. Clinical Variables: “Red Flags”

The latest evidence suggests that ATTRv should be suspected if progressive peripheral sensorimotor neuropathy is observed in combination with one or more of the following: autonomic dysfunction (erectile dysfunction, orthostatic hypotension, syncope), cardiomyopathy, gastrointestinal symptoms, unexplained weight loss, bilateral carpal tunnel syndrome (CTS), lumbar canal stenosis, renal impairment, ocular involvement (vitreous opacities), and/or family history of polyneuropathy, cardiomyopathy, or ATTRv [4]. According to this evidence, we evaluated the presence of such red flags through a detailed questionnaire.

### 2.4. Machine Learning Analysis

The XGBoost (XGB) classifier was trained to classify positive and negative *TTR* mutation patients. The XGB is a gradient boosting algorithm that uses several decision trees to create the final model [23]. The decision trees are constructed sequentially to improve the failures of the previously trained trees. In fact, the training process aims to minimize a loss function by adding weak decision tree learners. This method is called a boosting ensemble method and has been shown to improve model accuracy [24]. The XGBoost model is established as a standard to process tabular data and improve the performance over deep architectures [25]. In fact, several applications showed a high performance of the XGBoost model when using a dataset with a limited number of samples [26,27,28]. Eventually, the XGBoost was compared with other models, namely, support vector machine (SVM) with linear kernel, logistic regression (LR), and decision tree (DT).

Considering the limited number of cases in the dataset, the model performance was computed using a 20-repeated stratified 10-fold cross-validation, ensuring a correct estimation of the model generalization capability. The stratified setting guaranteed the balancing between the two groups for the training and test. Accuracy, area under the receiver operating characteristic (AUC-ROC), specificity, sensitivity, positive predictive value (PPV), and negative predictive value (NPV) were computed as metrics to evaluate model performance, reporting mean and standard deviation.

In addition, to evaluate the contribution of each red flag in the XGB model, the SHAP three explainer method was used as an explainable AI algorithm [29,30]. The SHAP algorithm exploits the computation of the Shapley values to assess the contribution of each feature to the model decision process. It is a post-hoc explanation algorithm;, that is, it is applied after the training of the machine learning model. In our case, it was applied to the trained XGBoost model to estimate the most impactful features in the predictive process. Explaining the prediction is mandatory in medical domains because the patterns a model discovers may be more important than its performance [30]. The SHAP method is established as a reference for model explanation, proving its effectiveness in different contexts [31,32,33]. For this reason, it allows us to estimate the most impactful features in the predictive process. 

## 3. Results

Data from 397 patients affected by polyneuropathy of undetermined etiology who underwent TTR genotyping were initially considered for study inclusion in the study period. In particular, 213 TTR-mutated subjects and 184 patients with negative genetic testing were included. However, after removing 120 first-degree family members, 93 mutated ATTRv probands (age 68 (32–87) years, 72 (77%) males) were included. Among patients with negative genetic testing, 96 patients (age 69 (52–82) years, 70 (73%) males) were selected. Patients with positive and negative genetic testing were age- and sex-matched. Hence, a cohort of 189 patients, 93 with a positive and 96 with a negative genetic test, was considered for the classification task. As reported in Table 1, the 93 screening-positive ATTRv patients presented more frequently with bilateral CTS and autonomic dysfunction, followed by ataxia, unexplained weight loss, and cardiomyopathy. Among ATTRv patients, the most frequent mutations encountered were Phe64Leu (52%), Val30Met (31%), Glu89Gln (5%), and Val122Ile (3%) (Table 1). 

The min–max approach was used to normalize in the range [0, 1] the age feature. The XGBoost classifier was trained using the gbtree booster, with 0.2 as the learning rate, 0.8 as the L2 regularization term, and 100 estimators. The other hyperparameters were maintained at the default values. The linear kernel was set for the SVM. The achieved model’s performances were reported in Table 2. The XGB outperforms the other algorithms, showing a higher AUROC, accuracy, and most importantly, a balanced sensitivity and specificity. Conversely, the LR and SVM resulted in being unable to generalize to negative samples, considering the large imbalance between the sensitivity and specificity. The results suggest the capability of the XGB model to fairly identify both positive and negative samples. In particular, an AUC-ROC of 0.752 ± 0.107, accuracy of 0.707 ± 0.101, sensitivity of 0.712 ± 0.147, specificity of 0.704 ± 0.150, NPV of 0.726 ± 0.118, and PPV of 0.711 ± 0.119 were computed. Figure 1 shows the AUC-ROC curve of the considered models, computed during the 20-repeated 10-fold cross-validation.

In addition, Figure 2 shows the SHAP beeswarm plot, in which the features are ordered by importance. The graph calculated through the Shapley values enables a clinical introspection of the model, restoring the predictive red flags of positive or negative genetics.

The SHAP beeswarm plot (Figure 2) shows that age, bilateral CTS, and autonomic dysfunction were similarly distributed in both screening-positive and negative patients (blue and red dots on both sides). This result was unexpected as CTS and autonomic dysfunction were the most frequent symptoms in ATTRv patients, being reported in 51% of cases (Table 1). Of interest, less frequent symptoms such as ataxia, unexplained weight loss, gastrointestinal symptoms, and cardiomyopathy were predictive of ATTRv (red dots on the right side). Conversely, ocular involvement, autoimmunity, diabetes, lumbar spinal stenosis, and renal involvement were associated with a negative genetic test (red dots on the left side).

## 4. Discussion

This study explores the role of a machine learning approach to identify reliable clinical factors which might be predictive of a positive genetic test. The principal purpose was to develop a systematic approach capable of guiding genetic testing in the context of general practitioners and neurologists who are not confident with ATTRv amyloidosis. Unfortunately, the multisystemic clinical presentation of ATTRv often makes it difficult to distinguish it from other conditions, thus causing a significant misdiagnosis [5,6]. Recent evidence has clearly shown the importance of an early diagnosis in such a fatal and disabling disease, especially when dealing with treatable disorders [4]. Hence, reliable and standardized diagnostic approaches are in demand [4]. However, as ATTRv is a heterogeneous disease caused by over 130 different mutations in the *TTR* gene, several peculiar phenotypes have been reported in non-endemic countries depending on specific genotypes and environmental factors [34]. Indeed, we aim to develop a simple guide for genetic testing that may be useful for clinicians. 

The use of ML methods enables the analysis of complex hidden patterns between data. For this reason, features that individually appear not significant (univariate analysis) can become predictive when aggregated with other features through ML models (multivariate analysis) [21]. However, traditional ML methods allow for the development of highly accurate models while not guaranteeing model transparency. Through XAI algorithms, it was possible to validate the model findings and compare them with the medical literature. The explainability toward their users is becoming a requirement these systems should satisfy [35] to implement user acceptance and control [36], and to face the ethical and legal aspects [36]. In fact, our XGBoost model achieved a promising performance for the detection of a positive biopsy, and important findings were clinically validated via the SHAP explanation. In particular, the analysis suggested that unexplained weight loss, cardiomyopathy, gastrointestinal disturbances, and ataxia are useful clinical features to detect ATTRv patients among patients presenting with polyneuropathy (Figure 2). These results are far along with considering these as the main clinical features in ATTRv patients, as, in the cohort examined, bilateral CTS associated with autonomic dysfunction and ataxia was the most frequent clinical picture, encountered in 50% of cases in the selected population. However, it should be noted that this main core of symptoms of ATTRv is similar to the ones generated by other causes of polyneuropathy, first of all diabetic polyneuropathy, which was quite frequent in our control group (Figure 2). Hence, we might interpret these “red flags” as a guide to raise suspicion of ATTRv in patients presenting with an undetermined polyneuropathy associated with a bilateral carpal tunnel syndrome. In addition, it should be considered that even if CTS is considered a main feature of ATTRv [37], presenting many years before other more severe features, CTS is also highly frequent in other etiologies [38,39]. Ataxia in ATTRv is an expression of a prominent sensory fiber involvement, which characterizes the early phase of ATTRv [40], and often precedes the more disabling motor damage [14,41]. Gastrointestinal involvement is frequent in ATTRv [42], and diarrhea, constipation, or weight loss may be present since the onset of the disease, even anticipating neurological symptoms [43]. In addition, recent evidence proved that gastrointestinal involvement may alter body composition with a good reversal after gene silencing [44]. Furthermore, gastrointestinal symptoms are insidious and can be misinterpreted, thus causing a relevant diagnostic delay [43,45]. In this study, weight loss and gastrointestinal symptoms showed a similar prevalence rate in *TTR*-mutated patients (37–45%), confirming their possible pathophysiological correlation [46,47]. Finally, cardiac involvement in ATTRv is common as it can represent clinical onset [6,48,49]. Moreover, Phe64Leu and Val30Met, amounting to over 80% of cases, are usually associated with mixed phenotypes [6,49,50].

Conversely, renal and ocular dysfunction, as well as lumbar canal stenosis, did not represent in our cohort a sensitive red flag. A first consideration is that lumbar canal stenosis, ocular involvement, and renal dysfunction were also frequent in patients with a negative genetic test, probably because the target population was quite aged (Table 1). Moreover, many patients also presented with a cataract which might have caused an underestimation of the vitreous opacities in this cohort. However, the high frequency of diabetes in the control group might have influenced our results considering the potential ocular and renal damage which is characteristic of diabetes [51,52,53]. In addition, autonomic dysfunction and CTS are frequently encountered in diabetic polyneuropathy [39,54]. Many studies have explored the differential diagnosis between ATTRv and diabetes, showing their similarities and differences [55]. However, our data come from a real-life experience of screening which might be a strength of the study if we consider that diabetes is the most common worldwide cause of acquired neuropathy: from this perspective, these results underline that cardiac (hypertrophic cardiomyopathy) and gastrointestinal involvement (unexplained weight loss, diarrhea, constipation) in an ataxic patient might represent the most sensitive red flags to the diagnosis of ATTRv, even in diabetic patients.

## 5. Limitations and Future Directions

Our study presents several limitations that should be addressed. The first limitation concerns the small size of the dataset. As a consequence, the study sample is not wide enough to draw conclusions on specific mutation-related phenotypes, but results are inferred considering ATTRv as a whole. In particular, it is well established that the generalization capabilities of ML models (also called data-driven models) are related to the availability of large amounts of data. In fact, to propose a fair performance, a cross-validation procedure was applied. However, there is a lack of a structured comparison with a statistical ordinary analysis for the complete validation of such an instrument. A further limitation comes from the concept of “red flag”, which can be self-reported by the patient, described in a specialist’s report, and demonstrated by an instrumental examination with different grades of precision in the clinical assessment. Hence, the assessment of such red flags might be poor and incomplete, due to underreporting or undervaluation (i.e., ocular and cardiac assessments, evaluation of erectile dysfunction). Hence, poor assessment as well as misdiagnosis with diabetes might explain the low predictive value of ocular and renal symptoms in this cohort. Finally, we included data from four specialized centers for the care of ATTRv from Palermo, Messina, Naples, and Rome; hence, our results might be reliable only when related to specific mutations from Italy.

## 6. Conclusions

Hereditary transthyretin amyloidosis with polyneuropathy (ATTRv) is an adult-onset multisystemic disabling and fatal disease, affecting the peripheral nerves, heart, gastrointestinal tract, eyes, and kidneys. Nowadays, several treatment options are available; thus, avoiding misdiagnosis is crucial to starting therapy in early disease stages. Our data support the use of ML and XAI algorithms in clinical screening to raise the suspicion of ATTRv, thus contributing to a potential reduction in the diagnostic delay in non-endemic areas. ATTRv should be suspected if progressive peripheral sensorimotor neuropathy is observed in combination with ataxia, gastrointestinal problems, unexplained weight loss, and cardiomyopathy. Further studies are needed to explore the clinical application of an ML algorithm in ATTRv.

## Figures and Tables

**Figure 1 brainsci-13-00805-f001:**
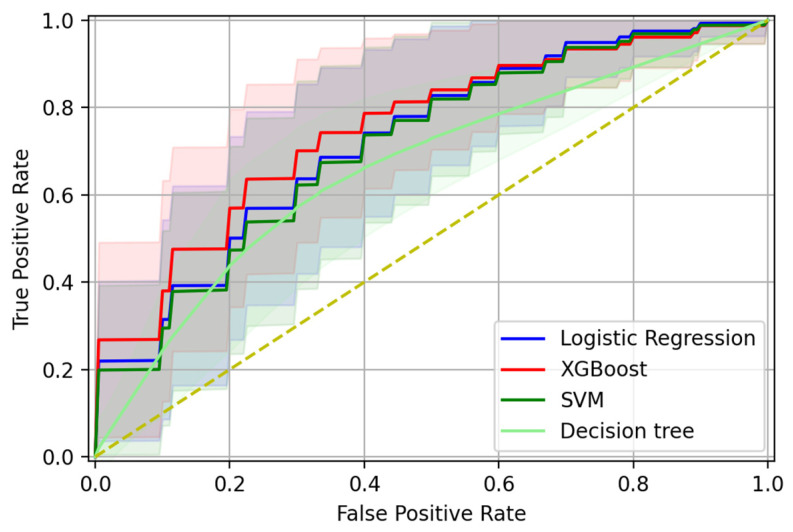
ROC curve of the models computed during the 20-repeated 10-fold cross-validation procedure. AUC-ROC, area under the receiver operating characteristic curve; ROC, receiver operating characteristic, SVM, Support Vector Machine.

**Figure 2 brainsci-13-00805-f002:**
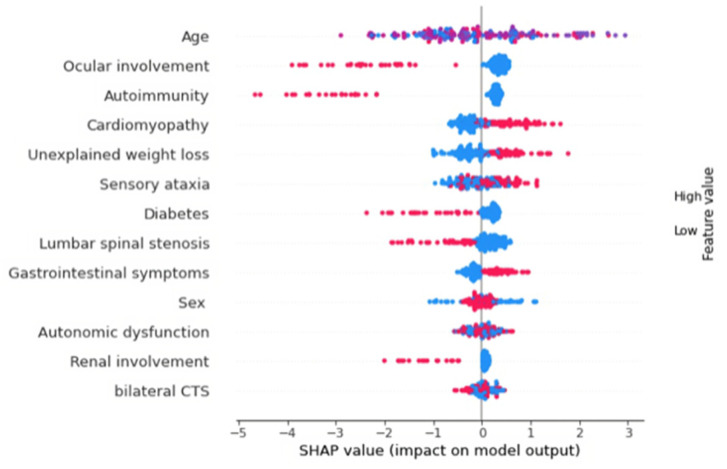
SHAP beeswarm plot. The red dots indicate the presence of the red flag, the blue dots indicate the absence of the red flag. For the age features, red dots mean older patients, blue dots younger. The red dots on the left of the graph indicate that the presence of the red flag is predictive of no TTR mutation, while those on the right are predictive of TTR mutation (same for blue dots). TTR, transthyretin.

**Table 1 brainsci-13-00805-t001:** Clinical features and genotypes encountered in the screened-positive ATTRv patients. ATTRv, Hereditary transthyretin amyloidosis with polyneuropathy; TTR, transthyretin; * Chi-square with *p* < 0.05.

Clinical Features	Screening-Positive ATTRv Patients (n = 93)	Screening-Negative Patients (n = 96)	*p* Value
Age (years)	68 (32–87)	69 (52–82)	0.24
Gender (males)	72 (77%)	70 (73%)	0.29
Bilateral carpal tunnel syndrome	47 (51%)	51 (53%)	0.42
Autonomic dysfunction	47 (51%)	50 (52%)	0.47
Ataxia	45 (48%)	46 (48%)	0.53
Unexplained weight loss	42 (45%)	30 (31%)	0.034 *
Cardiomyopathy	39 (42%)	35 (36%)	0.26
Gastrointestinal disturbances	34 (37%)	40 (42%)	0.28
Lumbar canal stenosis	19 (20%)	28 (26%)	0.11
Diabetes	7 (8%)	24 (25%)	0.001 *
Ocular disorders	5 (5%)	27 (28%)	<0.001 *
Renal dysfunction	4 (4%)	13 (14%)	0.023 *
Autoimmunity	2 (2%)	21 (22%)	<0.001 *
TTR Mutations			
Phe64Leu	48 (52%)	-	-
Val30Met	29 (31%)	-	-
Glu89Gln	5 (5%)	-	-
Val122Ile	3 (3%)	-	-
Others	1 (8%)	-	-

**Table 2 brainsci-13-00805-t002:** Achieved performance of the XGBoost, Linear Regression (LR), Support Vector Machine (SVM), and Decision Tree (DT) models, computed considering the 20-Repeated 10-Fold Cross-Validation. AUC-ROC, area under the receiver operating characteristic curve; PPV, positive predictive value; NPV, negative predictive value.

	Accuracy	AUC-ROC	Sensitivity	Specificity	PPV	NPV
XGBoost	0.707 ± 0.101	0.752 ± 0.107	0.712 ± 0.147	0.704 ± 0.150	0.711 ± 0.119	0.726 ± 0.118
LR	0.660 ± 0.099	0.725 ± 0.107	0.732 ± 0.135	0.592 ± 0.150	0.641 ± 0.102	0.703 ± 0.129
SVM	0.662 ± 0.099	0.713 ± 0.118	0.795 ± 0.165	0.534 ± 0.154	0.626 ± 0.095	0.749 ± 0.160
DT	0.656 ± 0.100	0.661 ± 0.101	0.644 ± 0.154	0.669 ± 0.143	0.660 ± 0.118	0.668 ± 0.114

## Data Availability

Data are available from the corresponding author upon a reasonable request.

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
