# Peer review of "Machine Learning for Early Diagnosis of ATTRv Amyloidosis in Non-Endemic Areas: A Multicenter Study from Italy"

_brainsci, 2023, doi:10.3390/brainsci13050805_

Round 1

Reviewer 1 Report

The paper presents the Machine-Learning for early diagnosis of ATTRv amyloidosis in non-endemic areas: a multicenter study from Italy. Overall, the scientific objective is important. The article is well written and comprehensive. There are no ethical concerns about this study. The research design is appropriate and the methods clearly explained. The interpretation of the results is clearly presented and adequately supported by the evidence adduced. The references are up-to-date and the most important studies have been cited.
There are some minor/major revisions needed. Please provide a point-by-point response to the following queries:

1. Please explain the abbreviations used in the tales and figures

2. Please provide a flowchart, which depicts the flow of information through the different phases enrolment of participants – how many were included and excluded, and the reasons for exclusions. 

3. Figure 2 is not sharp enough, please correct that

Author Response

Dear Editors and Reviewers

Thanks for your comments. We would like to submit our revised version of the manuscript for possible publication in Brain science.

#Reviewer 1

The paper presents the Machine-Learning for early diagnosis of ATTRv amyloidosis in non-endemic areas: a multicentre study from Italy. Overall, the scientific objective is important. The article is well written and comprehensive. There are no ethical concerns about this study. The research design is appropriate and the methods clearly explained. The interpretation of the results is clearly presented and adequately supported by the evidence adduced. The references are up-to-date and the most important studies have been cited.

A: we are very grateful for your appreciation and the precious suggestions.

There are some minor/major revisions needed. Please provide a point-by-point response to the following queries:

  1. Please explain the abbreviations used in the tales and figures

A: we added abbreviations in Table 1, Table 2, Figure 1, and Figure 2.

  1. Please provide a flowchart, which depicts the flow of information through the different phases enrolment of participants – how many were included and excluded, and the reasons for exclusions.

A: we thank the reviewer for this precious suggestion to improve our manuscript. We added this information in the results, as follows:

“Data from 397 patients affected by polyneuropathy of undetermined etiology who underwent TTR genotyping were initially considered for study inclusion in the study period. In particular, 213 TTR-mutated subjects and 184 patients with negative genetic testing have been included. However, after removing 120 first-degree family members, 93 mutated ATTRv probands (age 68 (32-87) years, 72 (77%) males) have been included. Among patients with negative genetic testing, 96 patients (age 69 (52-82) years, 70 (73%) males) have been selected.”

  1. Figure 2 is not sharp enough, please correct that

A: we thank the reviewer for this punctual suggestion. We replaced the figure with a high-quality picture.

Hoping in positive feedback we look forward to hearing from you soon.

Kind regards,

Filippo Brighina

Reviewer 2 Report

Authors presented the machine learning algorithm for assessing the Hereditary transthyretin amyloidosis with polyneuropathy, suggesting that ML could support for the better diagnosis of the disease.

This report will be of interest and will help other researchers in the field.

However, authors only present the encountered frequent mutations, the disease symptoms and predictions. Hence, it is not clear how the symptoms and the predictions could be caused from the molecular changes in TTR. Authors should discuss the connections between the mutations, changes in biomarkers and the symptoms. It is not easy to aging factors for the appearances of the symptoms, but authors need to try.

There are numerous minor grammar mistakes.

Author Response

Dear Editors and Reviewers

Thanks for your comments. We would like to submit our revised version of the manuscript for possible publication in Brain science.

#Reviewer 2

Authors presented the machine learning algorithm for assessing the Hereditary transthyretin amyloidosis with polyneuropathy, suggesting that ML could support for the better diagnosis of the disease.

This report will be of interest and will help other researchers in the field.

A: we are very grateful for your appreciation and the precious suggestions provided.

However, authors only present the encountered frequent mutations, the disease symptoms, and predictions. Hence, it is not clear how the symptoms and the predictions could be caused from the molecular changes in TTR. Authors should discuss the connections between the mutations, changes in biomarkers and the symptoms. It is not easy to aging factors for the appearances of the symptoms, but authors need to try.

A: we thank the reviewer for this precious consideration and the possibility to improve our manuscript.

Over 130 allelic variants have been described in the TTR gene, most of which are amyloidogenic. Clinical manifestations depend mainly on the existing variant. The great heterogeneity in penetrance data depend on phenotype, genotype, and environmental factors. Also, the penetrance of the disease in carriers of allelic variants in different areas is high variable, increasing the risk of suffering the disease with age. Val30Met is reported in early-onset ATTRv and most cases of late-onset ATTRv from Europe. On the contrary, very few reports examined other mutations in detail, but often they described small numbers of cases, or miscellanea of mutations all together. In the present study, the cohort consists of ATTRv patients from 4 Italian centres. Hence, genotypes and phenotypes examined are from the Centre and the South of Italy, a non-endemic region. The study sample is not wide enough to draw conclusion on specific mutation-related phenotypes. This is a relevant limitation of the study, and it has been discussed in the specific section.

“Our study presents several limitations that should be addressed. The first limitation concerns the small size of the dataset. As a consequence, the study sample is not wide enough to draw conclusions on specific mutation-related phenotypes, but results are inferred considering ATTRv as a whole.”

There are numerous minor grammar mistakes.

A: we thank the reviewer for this precious remark. We doublechecked the manuscript for grammar and style.

Hoping in positive feedback we look forward to hearing from you soon.

Kind regards,

Filippo Brighina

Reviewer 3 Report

This manuscript describes a new approach to identifying ATTR amyloidosis among individuals with peripheral neuropathy of unknown etiology. The authors use a machine learning approach to identify ATTR polyneuropathy based on each patients’ other symptoms. The goal is to identify patients who should undergo genetic testing.

This is a potentially interesting approach, but there are two significant flaws in the manuscript.

First, there is no validation cohort, so it is not clear whether the model is broadly applicable. This could potentially be bypassed because identifying sufficient cases of a rare disease is challenging. However, the second major problem is that the performance of the ML model is not compared to any other approaches. The clinical data appear to mainly comprise qualitative categories of accompanying symptoms, which could presumably be addressed using a much simpler generalized linear model. If the authors can demonstrate that their ML approach is superior to a standard regression model, then this study could be presented as a preliminary result with the next step being a validation cohort. However, in the absence of this comparison, the value of manuscript is unclear.

Another problem with the manuscript is that the rationale for using this particular set of ML tools is unclear. It is not clear what XGBoost and SHAP are, why they are different and why they are suitable for this dataset.

The sample of patients may also be biased by the inclusion of only those individuals who were referred for genetic testing. Individuals with clearer diagnoses may not be tested even if red flag symptoms are recorded.

The manuscript has many minor language errors, ranging from undefined abbreviations in the abstract to fragments of leftover Italian (eg, “positive e negative” in the abstract). Another round of proofreading would be useful. The author list has a dangling “and” at the end, which presumably needs to be moved.

Author Response

Dear Editors and Reviewers

Thanks for your comments. We would like to submit our revised version of the manuscript for possible publication in Brain science.

#Reviewer 3

This manuscript describes a new approach to identifying ATTR amyloidosis among individuals with peripheral neuropathy of unknown etiology. The authors use a machine learning approach to identify ATTR polyneuropathy based on each patients’ other symptoms. The goal is to identify patients who should undergo genetic testing.

This is a potentially interesting approach, but there are two significant flaws in the manuscript.

A: we are very grateful for your appreciation and the precious suggestions, as well as the possibility to improve our manuscript.

First, there is no validation cohort, so it is not clear whether the model is broadly applicable. This could potentially be bypassed because identifying sufficient cases of a rare disease is challenging.

A: we thank the reviewer for this precious remark. This is an exploratory study evaluating the potential of ML approach to assess the predictive role of red flags in the clinical suspicion of ATTRv. This study has not the purpose of validation, but only suggest a different approach to ordinary statistical instruments. We added this consideration in the limitation section.

However, the second major problem is that the performance of the ML model is not compared to any other approaches. The clinical data appear to mainly comprise qualitative categories of accompanying symptoms, which could presumably be addressed using a much simpler generalized linear model. If the authors can demonstrate that their ML approach is superior to a standard regression model, then this study could be presented as a preliminary result with the next step being a validation cohort. However, in the absence of this comparison, the value of manuscript is unclear.

A: we thank the reviewer for this relevant consideration. We agree with the reviewer regarding the limitation, that were clearly stated in the specific section. However, ML approach and standard regression model are different instruments that see and demonstrate different things. Indeed, it is possible to find associations with ML approach that are not significant with ordinary statistics. ML approach offers the possibility to develop data-driven results interacting each-other with a different approach respect to standard statistical hierarchical approach. Hence, in our opinion, a direct comparison, which might conduct to different results, is not useful or necessary.

Another problem with the manuscript is that the rationale for using this particular set of ML tools is unclear. It is not clear what XGBoost and SHAP are, why they are different and why they are suitable for this dataset.

A: we thank the reviewer for this comment. We added a detailed explanations of the techniques used and motivated their choice, as here reported:

“The XGB is a Gradient Boosting algorithm that uses several Decision Tree to create the final model(1). The Decision Trees are constructed sequentially to improve the failures of the previously trained trees. In fact, the training process aims to minimize a loss function by adding weak Decision Tree learners. This method is called boosting ensemble method and has been show improves model accuracy(2). The XGBoost model is established a standard to process tabular data and improves the performance over deep architectures(3).”

“The SHAP Tree Explainer is a post-hoc explanation algorithms, that is, it is applied after the training of the machine learning model. In our case, it was applied to the trained XGBoost model to estimate the most impactful features in the predictive process. Explaining the prediction is mandatory in medical domains because the patterns a model discovers may be more important than its performance(4). The SHAP method is established as a reference for model explanation, proving its effectiveness in different contexts(5–7).”

We also added the following references to better clarify the use of XGBoost and SHAP:

  1. Friedman JH. Stochastic gradient boosting. Comput Stat Data Anal. 2002 Feb 28;38(4):367–78.
  2. Hosni M, Abnane I, Idri A, Carrillo de Gea JM, Fernández Alemán JL. Reviewing ensemble classification methods in breast cancer. Comput Methods Programs Biomed [Internet]. 2019 Aug 1 [cited 2023 Apr 27];177:89–112. Available from: https://pubmed.ncbi.nlm.nih.gov/31319964/
  3. Shwartz-Ziv R, Armon A. Tabular Data: Deep Learning is Not All You Need. Information Fusion [Internet]. 2021 Jun 6 [cited 2023 Apr 27];81:84–90. Available from: https://arxiv.org/abs/2106.03253v2
  4. Lundberg SM, Erion G, Chen H, DeGrave A, Prutkin JM, Nair B, et al. From Local Explanations to Global Understanding with Explainable AI for Trees. Nat Mach Intell [Internet]. 2020 Jan 1 [cited 2023 Mar 19];2(1):56–67. Available from: https://pubmed.ncbi.nlm.nih.gov/32607472/
  5. Scheda R, Diciotti S. Explanations of Machine Learning Models in Repeated Nested Cross-Validation: An Application in Age Prediction Using Brain Complexity Features. Applied Sciences 2022, Vol 12, Page 6681 [Internet]. 2022 Jul 1 [cited 2023 Apr 27];12(13):6681. Available from: https://www.mdpi.com/2076-3417/12/13/6681/htm
  6. Wang J, Gribskov M. IRESpy: An XGBoost model for prediction of internal ribosome entry sites. BMC Bioinformatics [Internet]. 2019 Jul 30 [cited 2023 Apr 27];20(1):1–15. Available from: https://link.springer.com/articles/10.1186/s12859-019-2999-7
  7. Alves MA, Castro GZ, Oliveira BAS, Ferreira LA, Ramírez JA, Silva R, et al. Explaining machine learning based diagnosis of COVID-19 from routine blood tests with decision trees and criteria graphs. Comput Biol Med. 2021 May 1;132:104335.

The sample of patients may also be biased by the inclusion of only those individuals who were referred for genetic testing. Individuals with clearer diagnoses may not be tested even if red flag symptoms are recorded.

A: we thank the reviewer for this important consideration. However, we enrolled all consecutive patients referred for genetic testing, excluding their relatives. In our centres is a common practice to offer genetic testing in any case of ATTR, apart from the clinical picture to offer a broad choice for treatment. Indeed, Italian regulatory authority gives consent to drug prescription (gene silencers, ASO, or TTR stabilizers) only in the presence of a pathogenic mutation in the TTR gene for all drugs approved. Also, we can affirm that in all patients diagnosed with ATTRv genetic testing was performed in Italian Centres. Moreover, the presence of a point mutation in the TTR gene was the “model training measure” of the ML model.

The manuscript has many minor language errors, ranging from undefined abbreviations in the abstract to fragments of leftover Italian (eg, “positive e negative” in the abstract). Another round of proofreading would be useful. The author list has a dangling “and” at the end, which presumably needs to be moved.

A: we thank the reviewer for this precious remark. We doublechecked the manuscript for grammar mistakes and acronyms.

Hoping in positive feedback we look forward to hearing from you soon.

Kind regards,

Filippo Brighina

Reviewer 4 Report

The authors examined the usefulness of machine learning for the diagnosis of hereditary transthyretin (ATTRv) amyloidosis with polyneuropathy. By examining 189 patients suspected to have ATTRv amyloidosis, unexplained weight loss, gastrointestinal symptoms, and cardiomyopathy showed a significant association with the positive genetic diagnosis, while bilateral carpal tunnel syndrome, diabetes, autoimmunity, ocular involvement, and renal involvement were associated with a negative genetic test. 

This is an interesting study suggesting the usefulness of machine learning for the diagnosis of ATTRv amyloidosis. It provides important insights into the early diagnosis of this disease and will attract broad range of readers. Taking up the topic of ATTRv amyloidosis is timely because new therapeutic options for this disease, such as transthyretin stabilizers, RNA interfering agents, antisense oligonucleotides, and gene editing agents, now appear one after another. The manuscript is well written, and I do not have any critical comments.

Suggestions to strengthen this manuscript are raised as follows: 

1. Please reconfirm the use of abbreviations throughout the manuscript. For example, “CTS” should be spelled out at its first appearance in the abstract and main text. “CIDP” is not needed because it appears only once in the text. 

2. Is this a retrospective study, or a prospective one?

3. As for the description between ATTRv amyloidosis and chronic inflammatory demyelinating polyneuropathy, an earlier study detailed this issue (Amyloid 2011; 18: 53-62). I would suggest citing this study in the first paragraph of the introduction section. 

Fine. 

Author Response

Dear Editors and Reviewers

Thanks for your comments. We would like to submit our revised version of the manuscript for possible publication in Brain science.

#Reviewer 4

The authors examined the usefulness of machine learning for the diagnosis of hereditary transthyretin (ATTRv) amyloidosis with polyneuropathy. By examining 189 patients suspected to have ATTRv amyloidosis, unexplained weight loss, gastrointestinal symptoms, and cardiomyopathy showed a significant association with the positive genetic diagnosis, while bilateral carpal tunnel syndrome, diabetes, autoimmunity, ocular involvement, and renal involvement were associated with a negative genetic test.

This is an interesting study suggesting the usefulness of machine learning for the diagnosis of ATTRv amyloidosis. It provides important insights into the early diagnosis of this disease and will attract broad range of readers. Taking up the topic of ATTRv amyloidosis is timely because new therapeutic options for this disease, such as transthyretin stabilizers, RNA interfering agents, antisense oligonucleotides, and gene editing agents, now appear one after another. The manuscript is well written, and I do not have any critical comments.

A: we are very honoured for your appreciation and the precious suggestions provided.

Suggestions to strengthen this manuscript are raised as follows:

  1. Please reconfirm the use of abbreviations throughout the manuscript. For example, “CTS” should be spelled out at its first appearance in the abstract and main text. “CIDP” is not needed because it appears only once in the text.

A: we thank the reviewer for this punctual remark. We doublechecked the manuscript for the use of acronyms and revised mistakes.

  1. Is this a retrospective study, or a prospective one?

A: we thank the reviewer for this specification. This is a retrospective study. Indeed, we applied ML approach to a pre-existent dataset of patients selected in a multicentric cohort. We specified it better in the methods.

  1. As for the description between ATTRv amyloidosis and chronic inflammatory demyelinating polyneuropathy, an earlier study detailed this issue (Amyloid 2011; 18: 53-62). I would suggest citing this study in the first paragraph of the introduction section.

A: we thank the reviewer for this suggestion. We added this useful reference in the introduction in the following sentence:

“For example, chronic inflammatory demyelinating polyradiculoneuropathy, diabetes, sensory ataxia, and amyotrophic lateral sclerosis (ALS) are commonly overlapping with ATTRv (5–7).”.

Hoping in positive feedback we look forward to hearing from you soon.

Kind regards,

Filippo Brighina